# Heterocyclic Amine Formation in Grilled Chicken Depending on Body Parts and Treatment Conditions

**DOI:** 10.3390/molecules25071547

**Published:** 2020-03-28

**Authors:** Dániel Pleva, Katalin Lányi, Kitti Dóra Monori, Péter Laczay

**Affiliations:** Department of Food Hygiene, University of Veterinary Medicine, István 2, H-1078 Budapest, Hungary; lanyi.katalin@univet.hu (K.L.); kittimonori94@gmail.com (K.D.M.); laczay.peter@univet.hu (P.L.)

**Keywords:** heterocyclic amines, HAR, NOR, MeIQx, 4,8-DiMeIQx, PhIP, chicken meat, carcinogenicity

## Abstract

Heterocyclic amines (HCAs) carcinogenicity is known since the 1970′s, but the exact way of their formation is still unclear. During these examinations different body parts (breast filet with and without skin, thigh filet without skin and full wing with skin) of chickens from the same Ross 308 strain were analyzed after grilling with the combination of 3-3 temperature and duration levels (150-180-210 °C and 2.5-5-10 min per side). Five different kinds of heterocyclic amines (HAR, NOR, MeIQx, 4,8-DiMeIQx and PhIP) were detected by HLPC-MS/MS. The results obtained from the present study confirm that, in general, the higher the temperature and longer the duration of the grilling the more HCAs will be generated. Grilling of chicken thigh without bones and skin resulted in lower amounts of HCAs generated in comparison to the grilling of chicken breast without skin. The presence of skin on the chicken breast increased the amounts of HCAs formed, especially if grilling was performed at high temperature for longer duration, especially at 210 °C for 10 min. In case of grilling the chicken wings, the amounts of HCAs formed were lower than observed in the breast.

## 1. Introduction

Heterocyclic amines or HCAs are organic molecules with at least one aromatic ring containing a nitrogen atom [1]. There are two main groups of HCAs that have an effect on food safety, the thermic (IQ-type; aminoimidazoaroarenes) and the pirolytic (non-IQ-type; aminocarbolines) ones [2].

Their production and impact on human health is different. The IQ-type HCAs (such as 3,8-dimethyl-imidazo[4,5-f]quinoxaline—MeIQx, 2-amino-3,4,8-trimethyl-imidazo [4,5-f]quinoxaline—4,8-DiMeIQx or 2-amino-1-methyl-6-phenylimidazo[4,5-b]pyridine—PhIP) are created from creatine or creatinine, aminoacids and reducing sugar on lower temperature (150 °C) during the so called Maillard reaction [3]. These ingredients are available in muscles so these compounds are mostly characteristic to heat treated meat products. These molecules are on the World Health Organization’s International Agency on Research of Cancer’s (WHO-IARC) 2A or 2B list (probably or possibly carcinogenic to humans) [4]. The non-IQ-type HCAs (such as 1-methyl-9*H*-pyrido[3,4-*b*]indole—HAR or 9*H*-β-carboline—NOR) are produced due to the pyrolysis of proteins or aminoacids over 300 °C. They are not direct carcinogens, but they can strengthen the effect of carcinogenes so they are called co-carcinogenes [5].

The carcinogenicity of these molecules is based on their ability to create DNA-adducts. HCAs are absorbed from the gastrointestinal tract easily and then they reach the cytochrome p450 system [6]. CYP1A2 catalyzes the N-hydrolization and at the end of the reaction chain arilnitrenium ion is formed that can be adducted to the DNA. Animal experiments proved the fact that HCAs can cause several types of malign and benign tumors, most commonly in the large intestines and the liver [7,8].

The HCA profile depends on the species of the animal the meat derive from, also the body part, the temperature and the duration and the type of the heat treatment [9]. The IARC in 2015 classified red mammal meat to group 2A (probably carcinogenic) and processed meat to group 1 (carcinogenic to humans) partly because of their HCA content [10]. However, the components of HCAs are available in poultry meat as well [11] and previous bibliographic data confirmed the suspicion of the considerable HCA content of heat treated poultry meat products [12,13]. According to statistics poultry meat takes the highest ratio of meat consumption in several European countries [14], so a significant part of the population may be at risk. Surveys conducted by our department [15,16] show that although most of the respondents eat meat, they are not informed about the risk that they can cause by using an inappropriate cooking method and irresponsibly the temperature and the duration of the heat treatment are not observed carefully.

Although there are a number of bibliographic data in the scientific literature on the formation and profiles of HCAs, the data refer mostly to mammalian meat types [17,18], and less frequently to poultry meat [9,19]. In addition, the information available for poultry meat is incomplete in terms of main factors affecting the formation of HCAs, such as the body parts (e.g., breast, thigh, wings), the presence of skin, and the role of the temperature-time combination used for heat treatment.

Our aim was to compare different body parts of the same kind of chicken to reveal the differences between their HCA profile after the same heat treatment by detecting the five most frequent molecules (HAR, NOR, MeIQx, 4,8-DiMeIQx and PhIP) in poultry meat. In addition, we intended to get information on the impact of the temperature-time combination applied and on the role of the adhering skin in case of the most widely consumed chicken breast. These are important factors affecting the formation of the carcinogenic HCAs and therefore, important aspects of determining the least harmful, easiest and most easily verifiable methods of home cooking.

## 2. Results

### 2.1. Validation of the LC-MS/MS Method

Results obtained from validation of the LC-MS/MS method used for quantitation HCAs in grilled chicken parts are presented in Table 1.

As can be seen, all parameters met the requirements. Further information of the validation is available in the Materials and Methods.

### 2.2. HCA Content of Chicken Bodyparts

The chicken breasts without skin failed to show high HCA content when they were heat treated at 150 °C, not even if it took the longest period of treatment, i.e., 10 min both sides. Only, NOR was detectable at this temperature after 10 min of treatment. The 180 °C 2.5 min combination also seemed to be low for any significant HCA production. However, at a 10 min treatment at this temperature every single HCA became detectable, except for HAR. At 210 °C, every treatment duration was sufficient to produce all the detected HCAs, however their concentrations significantly depended from the time of grilling applied. The highest concentration was detected in case of PhIP, where 15.73 ng/g level was observed after 10 min that represented 75.0% of the all HCA content measured (Table 2).

The presence of skin on the chicken breast slices slightly changed the portfolio of the HCAs detected. For example, more HCAs were detected at 150 °C-10 min heat treatment, at 180 °C-10 min heat treatment and at 210 °C applying 5 or 10 min grilling time. However, at 210 °C-2.5 min grilling, MeIQx and 4,8-DiMeIQx were not measurable and the cumulate HCA level was also lower than in case of skinless chicken breast filets (Table 3).

In case of thigh samples, the appearance of HAR was more determinative, it was already quantifiable at 150 °C after 5 min of heat treatment. The quantity of other HCAs was usually lower compared to the breast meat treated by the same temperature and time. Especially, PhIP reached a lower level of 3.39 ng/g which is only 21.6% of its concentration in the chicken breast. In total, that made up only 53.7% of the total HCA amount, while the ratio of HAR was 9.0% (Table 4).

Chicken wings did not contain any HCAs in a detectable amount grilled at 150 °C (except for HAR at 10 min) and at 180 °C after 2.5 or 5 min of grilling. At 180 °C, the 10-min grilling resulted in measurable amount of all HCAs tested, with PhIP detected in the highest concentration. At 210 °C, the grilling for 2.5 min generated HAR and PhIP in detectable concentrations, whereas at 5 min and 10 min of grilling all other HCAs tested appeared (Table 5).

The total HCA content of the chicken breast filets with and without skin, the thigh filets without skin and the chicken wings that contain skin and bones, as well generated as a function of the grilling temperature and time is presented graphically on a logarithmic scale in Figure 1.

## 3. Discussion

In the present studies, chicken breast with or without the adhering skin, thigh and wings were grilled at different temperature and time combinations frequently used in the everyday home practice in order to detect the formation of carcinogenic HCAs as a function of the temperature-time combination of grilling and the affecting role of adhering skin. 

As it could be expected from the available bibliographic data [20], within one meat type, higher temperature and longer duration of heat treatment caused significant (*p* < 0.05) increase in the amount of each HCA generated. Kondjoyan et al. received a similar tendency in beef that was heat treated by another method [20]. In both cases, the duration and the temperature were found to have an important role. However, the heat treatment, i.e., the temperature and time combination of grilling also influences the sensory properties of the grilled chicken parts, and the efficacy of killing the microorganisms present on the surfaces or in the internal part of the meat. Therefore, these aspects and correlations will be investigated in further studies.

Comparing the results obtained from grilling the chicken breast without skin and thigh samples without bones by applying the same temperature and time combinations it can be seen that the breast contained more HCAs, in general (Table 5). The difference became more pronounced with increasing the temperature and extending the duration of heat treatment. At grilling at 210 °C for 10 min, the generated amounts of all HCAs tested were significantly (*p* < 0.05) higher in the chicken breast than in the chicken thigh, except HAR. This difference can be due to the higher protein content of the breast [21]. In case of HAR, the difference could be explained with the different amino acid profile of the breast and the thigh [22]. Our results are consistent with the available bibliographic data in the topic, although the composite temperature-grilling time spectrum we used is only partially covered [19,20].

In case of breasts grilled with or without skin, the skinless results were in general lower than those obtained from samples grilled with the adhering skin (Table 5). The most pronounced difference could be observed at 210 °C for 10 min grilling where it proved be statistically significant (*p* < 0.05) in all cases. This phenomenon could be due to the higher fat content of the skin that can increase the HCAs production by a frying effect [19]. However, the impact of the presence of skin on the elevated production of HCAs did not appear in every combination. In some cases, especially for shorter treating times, it was observed to have more a protecting effect. Iwasaki et al. also discovered this unequivocalness in respect of the presence of skin [19] and our results seems to confirm the complexity of this issue.

The measured HCAs in grilled chicken wings at different temperature and time combinations were lower than in the chicken breast with skin in case of most combinations (Table 5). The difference was statistically significant (*p* < 0.05). This could be due to the presence of bones or the lower protein content of the wings compared to the breast as described for roasted chicken body parts could be indicative of grilled parts, as well [23]. However, when assessing the wing results, probably the special shape of the wings should also be considered as a consequence of which the size of their surface touching the grill plate was lower than with the breast.

## 4. Materials and Methods 

### 4.1. Chemicals

Out of the HCA standards MeIQx, 4,8-DiMeIQx and PhIP were purchased from Toronto Research Chemicals (North York, ON, Canada), HAR and NOR from Sigma-Aldrich (Saint Loius, MO, USA), as well as caffeine used as internal standard. Other chemicals such as acetonitrile, dimethylformamide, methanol, formic acid, acetic acid and sodium hydroxide were purchased from VWR International (Radnor, PA, USA). Type I ultra-pure water was produced by SUEZ Environment^®^ Water Purification System (Paris, France).

### 4.2. Instruments and Equipment

The grilling was performed on an Electrolux ETG340 electric open contact grill (Electrolux, Stockholm, Sweden). For the sample preparation the Department’s Bosch hand blender, Certomat WR water tub shaker (Sartorius AG, Göttingen, Germany), Biofuge Primo R centrifuge (Kendro, Asheville, NC, USA) and Biotage VP evaporator (Biotage, Uppsala, Sweden) were used; the additional equipment was Phenomenex SI silicagel (Phenomenex, Torrance, CA, USA) and Phenomenex C18 columns (Phenomenex, Torrance, CA, USA). The analysis was carried out by a Shimadzu LCMS 8030 HPLC-MS/MS system (Shimadzu, Kioto, Japan) with a Phenomenex Kinetex C18 EVO 100 × 4.6 mm ID (2.6 µm particle size) column equipped with a 40 × 2 mm C18 guard column by LabSolutions^®^ software (Shimadzu, Kioto, Japan). Pure water was insured by the SUEZ Environment^®^ Water Purification System of the Department of Animal Hygiene (Budapest, Hungary).

### 4.3. Grilling Trials

The trials were carried out at the Department of Food Hygiene of the University of Veterinary Medicine, Budapest, Hungary. For the experiments, we used Ross 308 chicken meat (that was previously chemically analyzed by an accredited laboratory) obtained from retail market (Budapest, Hungary). Three chickens were sampled and three samples were taken from all the body parts of each animal in every combination of temperature and time.

The breasts and the thigh filets were cut into 40 g 1.6 cm thick slices (breasts both with or without the skin, thighs without skin), and wing samples were used in their original shape, covered with skin. For the heat treatment an Electrolux ETG340 electric open contact grill was used. The grilling conditions included nine combinations of three temperatures (150/180/210 °C) and three time periods (5/10/15 min) regarding both sides of the samples The applied grilling conditions were defined by consumers or commercial foodservice operations in the practice [24,25], the maximum capacity of the contact grill and the minimum temperature where we can expect the generation of HCAs.

### 4.4. Sample Preparation

Then 10 g were taken from each sample and were divided to five pieces of 2 g that were prepared separately. The samples were shredded by a Bosch hand blender (Bosch, Gerlingen, Germany), homogenized to start the saponification with 32 mL 1M NaOH solution. The mixture was shaken 190 rpm for 90 min on 60 °C by a water tub shaker (Certomat WR), then it was filled up to 50 mL with 1 M NaOH. The 10 mL aliquots were taken into centrifuge tubes and were centrifuged 8000 rpm for 10 min on 10 °C by a Biofuge Primo R centrifuge, then all the supernatant was taken onto Strata^®^ SI-1 Silica (55 µm, 70 Å), 500 mg/6 mL silicagel SPE columns. The columns were earlier conditioned by 2 mL H_2_O and 2 mL 1M NaOH-water solution. The sample was eluted by 2 × 1.5 mL ethyl acetate and then evaporated to dryness under N_2_ gas 50 °C in a Biotage VP evaporator. The rest of the NaOH mixture that had previously run down the SPE column was taken onto a Phenomenex C18 column that was pre-conditioned by 2 mL acetonitrile and 2 mL 1 M NaOH-water solution. The sample was eluted from the C18 by 2.5 mL acetonitrile into the same tube with the same numbered sample previously eluted and evaporated on silicagel, then it was evaporated again the same way. The dry sample was dissolved in 0.5 mL acetonitrile containing caffeine internal standard (150 ng/sample).

### 4.5. LC-MS/MS Analysis

Quantitative analysis of HCAs was carried out by a Shimadzu LCMS 8030 HPLC-MS/MS system. The chromatographic separation was done at a Phenomenex Kinetex C18 EVO 100x4.6 mm ID (2.6 µm particle size) column equipped with a 40 × 2 mm C18 guard column. Gradient elution was used with eluents ’A’: 50 mM ammonium acetate in water (pH 5 set by acetic acid) and ’B’ 0.1 V/V% formic acid in acetonitrile. The flow rate was 0.4 mL/min, a chromatographic run took 6 min. The column oven was set to 30 °C, the autoinjector’s temperature to 7 °C. The injected volume was 10 µl. The quadrupole tandem mass spectrometer was used with an electrospray ionization (ESI) ion source, in positive mode and multiple reaction monitoring (MRM). Other MS parameters were as follows: Interface 4.5 kV; interface temperature 250 °C; desolvation line 300 °C; heatblock 350 °C; detector 1.78 kV, nebulizing gas (N_2_) 3 L/min, drying gas (N_2_) 15 L/min; collision gas (Ar) 230 kPa.

Typical chromatograms obtained from analysis of the different body parts tested and the standards are presented in Figure 2.

### 4.6. Validation

Before starting the experiments, validation of the developed analytical method was carried out. Validation was performed in line with the requirements by the corresponding EU legislation and scientific guidelines [26,27]. Specificity/selectivity, linearity, limit of detection (LOD), limit of quantitation (LOQ), intra- and inter-day precision and recovery % were determined. Specificity of the method was examined by checking for the absence of interfering peaks in the appropriate mass spectrometric event (MRMs) at the expected retention times in a series of blank meat samples (treated at 70 °C for 20 min) subjected to the same sample preparation procedure. Linearity was evaluated by examining the calibration curves obtained by injecting solutions of known concentration. Individual calibration points were not allowed to differ by more than 15% from their nominal values (20% in the case of the lowest point of calibration). Limit of detection (LOD) was determined from the signal to noise (S/N) ratio of the MRMs in the case of analytical samples originating from blank meat samples in the time window of the given compound. Three times (3×) of S/N ratio was used as decision limit. Limit of quantitation (LOQ) was defined as the ng/g meat concentration of the given HCA calculated from the lowest point of the calibration curve that fits the validation parameters by the parameters of the sample preparation method.

Applicability of the method was determined by the intra- and inter-day precision and recovery %. Recovery % is expressed as the ratio of the measured concentration of a sample of known concentration and the nominal value (in percentage). Precision is expressed as the coefficient of variation (CV) of five parallel samples’ measured concentrations belonging to the same concentration. For intra-day parameters five samples were analyzed per concentration level within the same day, same batch For inter-day parameters five samples were analyzed per concentration level in three different days, in three different batches. For precision CV ≤ 15% was accepted. For recovery %, deviation of experimentally determined concentration values from the theoretical ones was accepted between −20% and +10%.

### 4.7. Data Processing and Statistical Analysis

Data processing was carried out first by the LabSolutions^®^ software of the LC-MS system. Secondary data processing and part of the statistical analysis were made by R statistical program (general linear mixed model - pairwise post hoc test), by MS Excel software and multi-way ANOVA calculation. Dependent variables of the models included the amounts of the HCAs (HAR, NOR, MeIQx, 4,8-DiMeIQx and PhIP), while the independent variables were the different kinds of meats (meat type and skin presence combined). The comparison was performed between the pairs presented in Table 5 where the temperature-time parameters were fixed between the pairs to point out the effect of the independent variables.

## 5. Conclusions

The results obtained from the present study confirm that, in general, the higher the temperature and longer the duration of the grilling, the more HCAs will be generated. 

Our results also reveal that the grilling of chicken thigh without bones and skin results in lower amounts of HCAs generated in comparison to the grilling of chicken breast without skin. The difference becomes more pronounced at higher temperature and longer duration of heat treatment.

The results obtained from the present studies also indicate that the presence of skin on the chicken breast may increase the amounts of HCAs formed, if grilling is performed for longer duration, especially at 210 °C for 10 min. However, when the time of grilling was short, the skin was found to decrease the formation of HCAs. Similarly, longer heat treatments at lower temperature reduced the formation of HCAs.

In case of grilling the chicken wings, the amounts of HCAs formed were lower than observed in the breast. However, due to the special shape of wings, the surface touching the grill plate was probably lower than in case of the breast that could influence the concentration of HCAs measured.

## Figures and Tables

**Figure 1 molecules-25-01547-f001:**
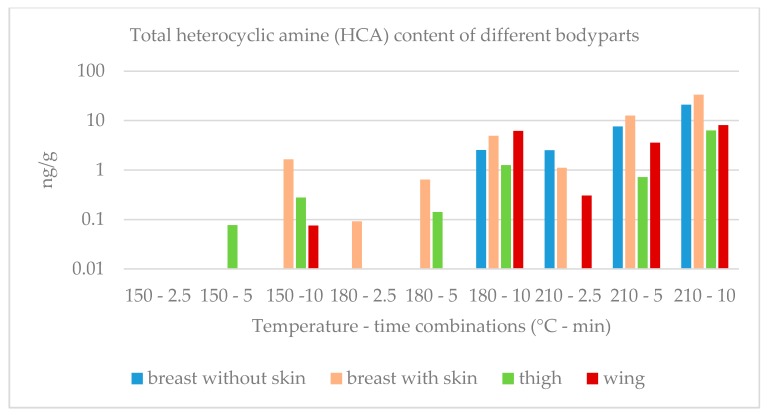
Total heterocyclic amine (HCA) content of chicken breast with and without skin, thigh and wing as a function of grilling temperature and time. (logarithmic scale) A: Breast without skin; B: Breast with skin; C: Thigh; D: Wing; E: Standard calibration.

**Figure 2 molecules-25-01547-f002:**
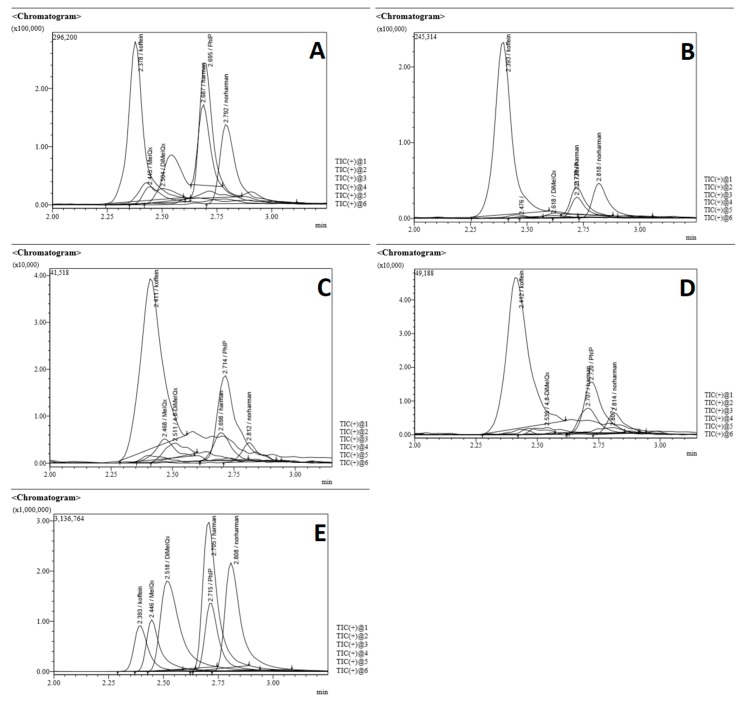
Chromatograms of heterocyclic amine (HCA) content of different chicken parts. A: Breast without skin; B: Breast with skin; C: Thigh; D: Wing; E: Standard calibration.

**Table 1 molecules-25-01547-t001:** Validation of the LC-MS/MS method used for quantitation HCAs in grilled chicken.

	Calibration Curve Parameters			LOD (ng·g^−1^)	LOQ (ng·g^−1^)	Precision		Recovery
Compound	Equation (y = a·x + b)*					Intra-Day	Inter-Day	
	a	b	r**			%	%	%
MeIQx	8 ± 1	0 ± 1E−3	0.9974 ± 0.0007	0.04	0.12	5 ± 1	7 ± 2	110 ± 10
4,8-DiMeIQx	11 ± 2	0 ± 2E−3	0.9965 ± 0.0002	0.02	0.08	5 ± 1	5.8 ± 0.9	110 ± 10
PhIP	7 ± 1	0 ± 3E−3	0.9982 ± 0.0005	0.03	0.12	4 ± 1	10 ± 2	112 ± 9
HAR	8 ± 1	0 ± 2.5E−3	0.9974 ± 0.0007	0.04	0.11	2.9 ± 0.8	6 ± 1	98 ± 8
NOR	8 ± 1	0 ± 2E−3	0.9973 ± 0.0003	0.05	0.17	4 ± 1	8 ± 2	100 ± 10

*y: The peak area ratio between the target compound and the internal standard at the given concentration level; *x: The ratio of concentrations; **r: Regression coefficient.

**Table 2 molecules-25-01547-t002:** Heterocyclic amine (HCA) content of heat-treated chicken breast without skin.

Breast Without Skin		HAR	NOR	MeIQx	4,8-DiMeIQx	PhIP	Total
T (°C)	t (min)	Amount (ng/g)					
150	2.5	<0.08	<0.08	<0.08	<0.08	<0.08	<0.08
	5	<0.08	<0.08	<0.08	<0.08	<0.08	<0.08
	10	<0.08	<0.08	<0.08	<0.08	<0.08	<0.08
180	2.5	<0.08	<0.08	<0.08	<0.08	<0.08	<0.08
	5	<0.08	0.13 ± 0.03 ^a^	0.09 ± 0.01 ^a^	<0.08	0.42 ± 0.06 ^a^	0.64 ± 0.06 ^a^
	10	<0.08	0.18 ± 0.02 ^b^	0.33 ± 0.05 ^b^	0.17 ± 0.02 ^a^	1.9 ± 0.3 ^b^	2.5 ± 0.3 ^b^
210	2.5	0.08 ± 0.02 ^a^	0.20 ± 0.03 ^b^	0.35 ± 0.06 ^b^	0.14 ± 0.02 ^b^	1.7 ± 0.2 ^b^	2.5 ± 0.2 ^b^
	5	0.20 ± 0.03 ^b^	0.47 ± 0.04 ^c^	1.0 ± 0.2 ^c^	0.49 ± 0.07 ^c^	5.5 ± 0.6 ^c^	7.6 ± 0.8 ^c^
	10	0.48 ± 0.05 ^c^	1.0 ± 0.1 ^d^	2.3 ± 0.2 ^d^	1.5 ± 0.2 ^d^	16 ± 2 ^d^	21 ± 2 ^d^

the lowest level of quantitation is 0.08 ng/g; T: Heating temperature; t: Duration of heat treatment; Values within a column not followed by the same superscript lowercase letter are significantly different (*p* < 0.05).

**Table 3 molecules-25-01547-t003:** Heterocyclic amine (HCA) content of heat-treated chicken breast with skin.

Breast With Skin		HAR	NOR	MeIQx	4,8-DiMeIQx	PhIP	Total HCA
T (°C)	t (min)	Amount (ng/g)					
150	2.5	<0.08	<0.08	<0.08	<0.08	<0.08	<0.08
	5	<0.08	<0.08	<0.08	<0.08	<0.08	<0.08
	10	0.19 ± 0.3 ^a^	0.32 ± 0.06 ^a^	<0.08	0.09 ± 0.01 ^a^	1.0 ± 0.1 ^a^	1.6 ± 0.2 ^a^
180	2.5	<0.08	0.09 ± 0.01 ^b^	<0.08	<0.08	<0.08	0.09 ± 0.01 ^b^
	5	0.30 ± 0.04 ^b^	0.15 ± 0.02 ^c^	<0.08	<0.08	0.20 ± 0.02 ^b^	0.64 ± 0.04 ^c^
	10	0.7 ± 0.1 ^c^	0.33 ± 0.04 ^a^	0.57 ± 0.07 ^a^	0.35 ± 0.04 ^b^	2.9 ± 0.3 ^c^	4.9 ± 0.3 ^d^
210	2.5	0.09 ± 0.01 ^d^	0.16 ± 0.02 ^d^	<0.08	<0.08	0.9 ± 0.1 ^d^	1.1 ± 0.1 ^e^
	5	0.39 ± 0.05 ^e^	0.51 ± 0.07 ^e^	0.7 ± 0.1 ^b^	0.49 ± 0.07 ^c^	10 ± 1 ^e^	12.6 ± 1.5 ^f^
	10	1.1 ± 0.2 ^f^	1.4 ± 0.2 ^f^	8.0 ± 0.9 ^c^	2.3 ± 0.3 ^d^	20.5 ± 2.5 ^f^	33 ± 4 ^g^

the lowest level of quantitation is 0.08 ng/g; T: Heating temperature; t: Duration of heat treatment; Values within a column not followed by the same superscript lowercase letter are significantly different (*p* < 0.05).

**Table 4 molecules-25-01547-t004:** Heterocyclic amine (HCA) content of heat-treated chicken thigh without skin and bones.

Thigh Without Skin		HAR	NOR	MeIQx	4,8-DiMeIQx	PhIP	Total HCA
T (°C)	t (min)	Amount (ng/g)					
150	2.5	<0.08	<0.08	<0.08	<0.08	<0.08	<0.08
	5	0.08 ± 0.01 ^a^	<0.08	<0.08	<0.08	<0.08	0.08 ± 0.01 ^a^
	10	0.17 ± 0.02 ^b^	0.11 ± 0.02 ^a^	<0.08	<0.08	<0.08	0.28 ± 0.02 ^b^
180	2.5	<0.08	<0.08	<0.08	<0.08	<0.08	<0.08
	5	0.14 ± 0.01 ^c^	<0.08	<0.08	<0.08	<0.08	0.14 ± 0.01 ^c^
	10	0.19 ± 0.03 ^d^	0.17 ± 0.02 ^b^	0.38 ± 0.05 ^a^	0.11 ± 0.02 ^a^	0.41 ± 0.05 ^a^	1.3 ± 0.1 ^d^
210	2.5	<0.08	<0.08	<0.08	<0.08	<0.08	<0.08
	5	0.17 ± 0.03 ^b,d^	0.13 ± 0.02 ^c^	<0.08	<0.08	0.41 ± 0.05 ^a^	0.71 ± 0.06 ^e^
	10	0.57 ± 0.06 ^e^	0.54 ± 0.07 ^d^	1.2 ± 0.2 ^b^	0.66 ± 0.07 ^b^	3.4 ± 0.4 ^b^	6.3 ± 0.6 ^f^

the lowest level of quantitation is 0.08 ng/g; T: heating temperature; t: duration of heat treatment; Values within a column not followed by the same superscript lowercase letter are significantly different (*p* < 0.05).

**Table 5 molecules-25-01547-t005:** Heterocyclic amine (HCA) content of heat-treated chicken wing with skin.

Wing With Skin		HAR	NOR	MeIQx	4,8-DiMeIQx	PhIP	Total HCA
T (°C)	t (min)	Amount (ng/g)					
150	2.5	<0.08	<0.08	<0.08	<0.08	<0.08	<0.08
	5	<0.08	<0.08	<0.08	<0.08	<0.08	<0.08
	10	0.08 ± 0.01 ^a^	<0.08	<0.08	<0.08	<0.08	0.08 ± 0.01 ^a^
180	2.5	<0.08	<0.08	<0.08	<0.08	<0.08	<0.08
	5	<0.08	<0.08	<0.08	<0.08	<0.08	<0.08
	10	0.38 ± 0.05 ^b^	0.34 ± 0.05 ^a^	0.8 ± 0.1 ^a^	0.27 ± 0.04 ^a^	3.9 ± 0.5 ^a^	6.2 ± 0.7 ^b^
210	2.5	0.08 ± 0.01 ^a^	<0.08	<0.08	<0.08	0.23 ± 0.03 ^b^	0.30 ± 0.04 ^c^
	5	0.81 ± 0.05 ^c^	0.36 ± 0.04 ^a^	0.35 ± 0.05 ^b^	<0.08	2.0 ± 0.3 ^c^	3.6 ± 0.4 ^d^
	10	0.9 ± 0.1 ^d^	0.8 ± 0.1 ^b^	0.91 ± 0.09 ^c^	0.36 ± 0.04 ^b^	4.5 ± 0.4 ^d^	8.1 ± 0.6 ^e^

the lowest level of quantitation is 0.08 ng/g; T: Heating temperature; t: Duration of heat treatment; Values within a column not followed by the same superscript lowercase letter are significantly different (*p* < 0.05).

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
