# Peer review of "Heterocyclic Amine Formation in Grilled Chicken Depending on Body Parts and Treatment Conditions"

_molecules, 2020, doi:10.3390/molecules25071547_

Round 1
Reviewer 1 Report
It is the third time that I review this paper, and in the previous two rounds I have highlighted the intrinsic incoherence in the significant figures between the average data and the relevant standard deviation reported in the tables. This is an utmost point in a paper dealing food analysis, and I am really disappointed observing that Authors have not yet corrected the data in the tables. For the last time I warmly suggest that Authors should provide to harmonize the mean value and the relevant standard deviation in ALL data reported in the tables. Unless this correction, I shall always against the publication of this paper in Molecules
Author Response
Dear Reviewer,
we would like to thank you for the extra effort you added to our paper. Now your insights are clear and I - as a beginner in the academic world - will keep in mind this presentation of data.
Yours faithfully,
Dániel Pleva
Reviewer 2 Report
188-189 "3 parallel samples" does not specify if the parts were from three different chickens.
194-195 The Lányi et al. 2018 poster at the 3rd Asian Sensory and Consumer Research Symposium only indicates the general types of cookery by men and women who consume meat and does not give any substantiation for the times and temperatures of grilling used for the samples in this research.
257-283 The conclusion section should summarize the findings in no more than 4 or 5 sentences and relate how the findings are important to science and to the application of the findings, all in a single paragraph. Most of the information currently in the conclusions section can be moved into the discussion section. Lines 257-258, 260-263, and 278-283 would be appropriate as sentences in the conclusions.
Author Response
Dear Reviewer,
thank you very much for your insights. They all pointed at important changes, that we have done the following way:
188-189 "3 parallel samples" does not specify if the parts were from three different chickens.
- Thank you for the comment, we have clarified the number of samples
194-195 The Lányi et al. 2018 poster at the 3rd Asian Sensory and Consumer Research Symposium only indicates the general types of cookery by men and women who consume meat and does not give any substantiation for the times and temperatures of grilling used for the samples in this research.
- Thank you for the comment, we have cited a study about grilling conditions
257-283 The conclusion section should summarize the findings in no more than 4 or 5 sentences and relate how the findings are important to science and to the application of the findings, all in a single paragraph. Most of the information currently in the conclusions section can be moved into the discussion section. Lines 257-258, 260-263, and 278-283 would be appropriate as sentences in the conclusions.
- Thank you for the remarks, we have removed the repetition of the discussion from the Conclusions section and restructured the Conclusions.
We hope that our changes satisfy your points of view.
Yours faithfully,
Dániel Pleva
Round 2
Reviewer 1 Report
Authors have been kept into account all my suggestions. Hence, in my opinion the paper is now publishable
Author Response
Dear Reviewer,
thank you very much for your comments, it really helped us to correct our article.
Yours faithfully,
Dániel Pleva
This manuscript is a resubmission of an earlier submission. The following is a list of the peer review reports and author responses from that submission.
Round 1
Reviewer 1 Report
The comments and suggestions for improve the quality of the paper have been reported as highlighted text and sticky notes in the pdf copy of the paper enclosed

Author Response
Dear Reviewer,
thank you very much for your help to make our article as good as possible. And having a PDF about the adviced corrections made it much easier to elaborate.
I attached your file completed with our points of view.
Thank you one more!
Yours faithfully,
Dániel Pleva

Reviewer 2 Report
The manuscript by Pleva et al reveals correlation between heterocyclic amines (HCAs) concentration in meat and selected body parts of chicken and grilling conditions. The authors investigated several parts of chicken meat: breast with and without skin, thigh without skin and bones, and wing with skin. They determined concentrations of 5 selected HCAs based on LC-MS/MS analyses.
The manuscript is written correctly, however a few details should be elucidated by authors:
i) Results section / Table 4. The authors write that DiMeIQx did not appear in probes of grilled meat at 5 min/210 C. However, the level of DiMeIQx at these conditions is reached 0.35 ng/g in the table 4.
ii) Section 4.5. Please, provide details of statistical analysis. The authors used one-way ANOVA analysis to compare two groups??? T-Student test would be sufficient for such analysis. Additionally, what post-hoc tests were used for ANOVA?
iii) Discussion and Conclusions. In my opinion, these parts should be re-written. There are no conclusions or hypotheses about the potential correlation of cooking conditions with population health. There is only inadequate discussion with available literature data. I think that after such changes the value of the manuscript would increase significantly.
Author Response
Dear Reviewer,
thank you very much for your help to make our article as good as possible.
Our resposes to your comments:
Thank you for the comment. We have corrected it We have re-checked the statistical analysis with a statistician. Thank you for the comment. We have changed it.Thank you once more!
Yours faithfully,
Dániel Pleva
Reviewer 3 Report
There are some wording and language and difficulties that must be corrected before acceptance of the manuscript for publication. The research adds to the body of information on HCA formation in poultry meat, but the manuscript has some errors that must be corrected and missing information that is necessary for other scientists to duplicate the experimental conditions.
Line(s) Comment
44 “meat” instead of “meal” since now many meals do not contain meat?
48-49 Clarification should be made by giving the reference of your previous results as the research in references 11 and 12 do not seem to have been conducted by any of the authors of this paper.
52 “respondents” instead of “answerers”
60 It was expected that the experimental conditions of the temperatures and times used in this study would be validated by references on the temperatures and times used by consumers or commercial foodservice operations to cook chicken.
89-105 Tables 1, 2 , 3, and 4 do not show the, if any, statistical differences among the heat treatments for any given HCA. Each table and figure should be independent of having to refer to any text so the titles should be “Heterocyclic amine (HCA) content . . . “
110 It must be established with a reference here, in the introduction, or in the materials and methods that the temperature and time combinations used in this study are representative of those used in every day home cooking of chicken.
124 It is expected that some of the references will be cited here.
132-133 This sentence is scientifically incorrect since neither breast without skin or wing had detectable HCA at two of the temperature and time combinations (150C for 2.5 and 5 min).
133-134 The referenced data from the National Chicken Council is for roasted chicken, not grilled chicken, and is for proximate analysis, not for HCA content.
134-136 This must be stated as a speculation rather than a fact unless the actual surface touching the grill plate was measured for the samples and could be compared with the weight or mass of the sample being cooked.
139 Figure 1 should indicate the differences between all treatment combinations (3 temperatures x 3 times x 2 breast types).
140 The total HCA contents of the 4 different parts that were analyzed could be shown in one figure rather than 3 figures since each figure contains the same data that is shown in one or more of the 3 figures.
155 The number of chickens that were sampled should be given.
160 “middle heat settings” instead of “middle degree”
188 The number of replications of the experimental conditions should be given. Since the Discussion section referred to significant differences among treatments, the method of determining the significance must be described.
190 The experimental design and statistical model should be stated as it is more appropriate to use a multi-factor ANOVA since 3 temperature by 3 time combinations (l. 161) x n replications were used.
209-210 As previously stated, this is speculative unless data on the surface area touching the grill plate compared to the sample weight and mass was obtained for each of the chicken parts and temperature-time combinations.
Author Response
Dear Reviewer,
thank you very much for your help to make our article as good as possible. We have made more corrections according to your comments.
44 “meat” instead of “meal” since now many meals do not contain meat? Thank you for the comment. The wording was corrected accordingly.
48-49 Clarification should be made by giving the reference of your previous results as the research in references 11 and 12 do not seem to have been conducted by any of the authors of this paper.
We have still added a reference that is more specific relating to the issue concerned.
52 “respondents” instead of “answerers”
We have amended accordingly.
60 It was expected that the experimental conditions of the temperatures and times used in this study would be validated by references on the temperatures and times used by consumers or commercial foodservice operations to cook chicken.
Thank you for this comment, we have added a new section about it.
89-105 Tables 1, 2 , 3, and 4 do not show the, if any, statistical differences among the heat treatments for any given HCA. Each table and figure should be independent of having to refer to any text so the titles should be “Heterocyclic amine (HCA) content . . . “
Thank you for the comment, we have signed the statistical differences.
110 It must be established with a reference here, in the introduction, or in the materials and methods that the temperature and time combinations used in this study are representative of those used in every day home cooking of chicken.
Thank you for the comment according to which we have included the missing information.
124 It is expected that some of the references will be cited here.
Thank you for the comment based on which we have completed the references.
132-133 This sentence is scientifically incorrect since neither breast without skin or wing had detectable HCA at two of the temperature and time combinations (150C for 2.5 and 5 min).
The sentence has been reworded. Please see the updated text.
133-134 The referenced data from the National Chicken Council is for roasted chicken, not grilled chicken, and is for proximate analysis, not for HCA content. Thank you for the comment, in this reference we would have liked to refer to the raw meat data
134-136 This must be stated as a speculation rather than a fact unless the actual surface touching the grill plate was measured for the samples and could be compared with the weight or mass of the sample being cooked.
Thank you for the comment based on which we have amended the wording.
139 Figure 1 should indicate the differences between all treatment combinations (3 temperatures x 3 times x 2 breast types).
Figure 1 is about skinless breast and thigh, that’s why there are no other breast data here
140 The total HCA contents of the 4 different parts that were analyzed could be shown in one figure rather than 3 figures since each figure contains the same data that is shown in one or more of the 3 figures.
Thank you for the comment. We also tried to combine the data in one figure, however we would think that maintaining three different figures can present the differences between the coherent pairs more legibly (Figure 1: bodyparts without skin; Figure 2:. role of the presence of skin; Figure 3:bodyparts with skin).
155 The number of chickens that were sampled should be given.
Thank you for the comment. The section Materials and Methods has been completed accordingly.
160 “middle heat settings” instead of “middle degree”
This sentence has been reworded.
188 The number of replications of the experimental conditions should be given. Since the Discussion section referred to significant differences among treatments, the method of determining the significance must be described. Description of the sample preparation has been completed and the method of determining the significance has bee added.
190 The experimental design and statistical model should be stated as it is more appropriate to use a multi-factor ANOVA since 3 temperature by 3 time combinations (l. 161) x n replications were used.
The multi-combinations were analysed by a general linear mixed model - pairwise post hoc test, the figures were examined by ANOVA.
209-210 As previously stated, this is speculative unless data on the surface area touching the grill plate compared to the sample weight and mass was obtained for each of the chicken parts and temperature-time combinations.
We have reworded the corresponding sentence considering the speculative aspect of this part of conclusion.
Thank you once more!
Yours faithfully,
Dániel Pleva

Round 2
Reviewer 1 Report
Many of the suggestions reported in the first round of the reviewing process have been taken into account by the authors. However, still one major criticism remains untouched: the intrinsic incoherence in the significant figures between the average data and the relevant standard deviation reported in the tables. The correct way in which the data is reported is not a questionable question: the data entered in the tables are still written incorrectly and - in this form -they cannot be published on Molecules
Reviewer 3 Report
It is disappointing that the authors dismissed most of the peer review comments as unimportant or inconsequential and so have diminished the scientific value of their research efforts.
Line(s) Comment
60-189 Nowhere in the manuscript are the conditions of the temperatures and times used in this study validated by references on the temperatures and times used by consumers or commercial foodservice operations to cook chicken. Simply stating that the grilling conditions were defined by consumers or commercial foodservice operations is not acceptable without supporting documentation.
95-160 Tables 1, 2 , 3, and 4 and Figures 1, 2 and 3 should be independent of relying on the text for understanding of abbreviations and acronyms so the titles should be “Heterocyclic amine (HCA) content . . . “
116-160 Much of this information, particularly the information in the figures, belongs in the results section. The Discussion section should explain scientifically why the results occurred and their meaning.
146-160 Confusion has been created in the meaning of the data in figures 1, 2, and 3 since there are two charts for each figure with the same abscissa and ordinate labels, but different data within the two charts without any additional explanation. It is not clear why values for one part and no value (or non detectable value) for the other part are not also indicated as being significantly different. Also, it is not clear why the total HCA contents of the 4 different parts that were analyzed could not be shown in one figure rather than 3 figures since each figure contains the same data that is shown in one or more of the 3 figures.
179 The number of chickens that were sampled should be given to define the inference space of the samples.
245-247 The experimental design and statistical model should be stated as it is more appropriate to use a multi-factor ANOVA since 3 temperature by 3 time combinations (l. 161) x n replications were used.
250-256, 261-274 This information restates the information in the results and discussion sections and should be deleted.